# Baby Intuitions Benchmark (BIB): Discerning the goals, preferences, and actions of others

**Kanishk Gandhi**
New York University

**Gala Stojnic**
New York University

**Brenden M. Lake**
New York University

**Moira R. Dillon**
New York University

## Abstract

To achieve human-like common sense about everyday life, machine learning systems must understand and reason about the goals, preferences, and actions of other agents in the environment. By the end of their first year of life, human infants intuitively achieve such common sense, and these cognitive achievements lay the foundation for humans' rich and complex understanding of the mental states of others. Can machines achieve generalizable, commonsense reasoning about other agents like human infants? The Baby Intuitions Benchmark (BIB)[1] challenges machines to predict the plausibility of an agent's behavior based on the underlying causes of its actions. Because BIB's content and paradigm are adopted from developmental cognitive science, BIB allows for direct comparison between human and machine performance. Nevertheless, recently proposed, deep-learning-based agency reasoning models fail to show infant-like reasoning, leaving BIB an open challenge.

## 1 Introduction

Humans have a rich capacity to infer the underlying intentions of others by observing their actions. For example, when we watch the animations from Heider and Simmel (1944) (see video[2] and Figure 1), we attribute goals and preferences to simple 2D shapes moving in a flat world. Using behavioral experiments, developmental cognitive scientists have found that even young infants infer intentionality in the actions of other agents. Infants expect agents: to have object-based goals (Gergely et al., 1995; Luo, 2011; Song et al., 2005; Woodward, 1998, 1999; Woodward and Sommerville, 2000); to have goals that reflect preferences (Repacholi and Gopnik, 1997; Kuhlmeier et al., 2003; Buresh and Woodward, 2007); to engage in instrumental actions that bring about goals (Carpenter et al., 2005; Elsner et al., 2007; Hernik and Csibra, 2015; Gerson et al., 2015; Saxe et al., 2007; Woodward and Sommerville, 2000); and to act efficiently towards goals (Gergely et al., 1995; Gergely and Csibra, 1997, 2003; Liu et al., 2019, 2017; Colomer et al., 2020).

Figure 1: A still from Heider and Simmel (1944). Despite the simplicity of the visual display, we ascribe intentionality to the three shapes in the scene: The large triangle is chasing the small triangle and the circle, whose goals are to avoid it.

Machine-learning and AI systems, in contrast, are much more limited in their understanding of other agents. They typically aim only to predict outcomes of interest (e.g., churn, clicks, likes, etc.) rather than to learn about the goals and preferences that underlie such outcomes. This impoverished "machine theory of mind"[3] may be a critical difference between human and machine intelligence more generally, and addressing it is crucial if machine learning aims to achieve the flexibility of human commonsense reasoning (Lake et al., 2017).

---

[1]The dataset and code are available here: https://kanishkgandhi.com/bib
[2]https://www.youtube.com/watch?v=VTNmLt7QX8E

35th Conference on Neural Information Processing Systems (NeurIPS 2021).

Recent computational work has aimed to focus on such reasoning by adopting several approaches. Inverse reinforcement learning (Ng et al., 2000; Abbeel and Ng, 2004; Ziebart et al., 2008; Ho and Ermon, 2016; Xu et al., 2019) and Bayesian approaches (Ullman et al., 2009; Baker et al., 2009, 2011, 2017; Jara-Ettinger, 2019) have modeled other agents as rational, yet noisy, planners. In these models, rationality serves as the tool by which to infer the underlying intentions that best explain an agent's observed behavior. Game theoretic models have aimed to capture an opponent's thought processes in multi-agent interactive scenarios (see survey: Albrecht and Stone, 2018), and learning-based neural network approaches have focused on learning predictive models of other agents' latent mental states, either through structured architectures that encourage mental-state representations (Rabinowitz et al., 2018) or through the explicit modeling of other agents' mental states using a different agent's forward model (Raileanu et al., 2018). Despite the increasing sophistication of models that focus on reasoning about agents, they have not been evaluated or compared using a comprehensive benchmark that captures the generalizability of human reasoning about agents. For example, some evaluations of machines' reasoning about agents have provided fewer than 100 sample episodes (Baker et al., 2009, 2011, 2017), making it infeasible to evaluate learning-based approaches that require substantial training. Other evaluations have used the same distribution of episodes for both training and test (Rabinowitz et al., 2018), making it difficult to measure how abstract or flexible a model's performance is. Moreover, existing evaluations have not been translatable to the behavioral paradigms that test infant cognition, and so their results cannot be analyzed in terms of the representations and processes that support successful human reasoning.

Our benchmark, the Baby Intuitions Benchmark (BIB), presents a comprehensive set of evaluations of commonsense reasoning about agents suitable for machines and infants alike. BIB adapts experimental stimuli from studies with infants that have captured the content and abstract nature of their knowledge (Baillargeon et al., 2016; Banaji and Gelman, 2013). It provides a substantial amount of training episodes in addition to out-of-distribution test episodes. Moreover, BIB adopts a "violation of expectation" (VOE) paradigm (similar to Riochet et al. (2018); Smith et al. (2019)), commonly used in infant research, which makes its direct validation with infants possible and its results interpretable in terms of human performance. The VOE paradigm, moreover, offers an additional advantage relative to other measures of machine performance like predictive accuracy, in that it reveals how an observer might fail: VOE directly contrasts one outcome, which requires a high-level, human-like understanding of an event, to another one, which instantiates some lower-level, heuristically, or perceptually based alternative. BIB thus presents both a general framework for designing any benchmark aiming to examine commonsense reasoning across domains as varied as agents, objects, and places, as well as a key step in bridging machines' impoverished understanding of intentionality with humans' rich one.

AGENT (Shu et al., 2021), a benchmark developed contemporaneously with BIB, is inspired by infants' knowledge about agents and has been validated with behavioral data from adults. Similar to BIB, AGENT challenges machines to reason about the intentions of agents as the underlying cause of their actions. Both benchmarks test whether models can predict that agents have object-based goals and move efficiently to those goals. There are nevertheless key differences between BIB and AGENT. First, BIB evaluates whether models can reason about multiple agents, inaccessible goals, instrumental actions, and the differences between the intentions of rational and irrational agents; AGENT does not test these competencies. The advantage of including them is that they introduce additional elements—beyond a single rational agent and one or two possible goal objects—that models must flexibly account for in their reasoning. These competencies, moreover, extend the infant cognition literature, potentially allowing BIB to further inform tests for infants. Second, BIB and AGENT evaluate new models differently. AGENT involves training on many different leave-out splits, where those splits include relatively minor differences between the training and test sets. BIB, in contrast, presents a single canonical split designed to maximally evaluate the abstractness of a model's reasoning: Models tested on BIB must flexibly combine learning from different types of training scenarios to solve a novel test scenario. We thus ultimately see BIB and AGENT as complementary and hope that new models focused on commonsense reasoning about agents will be evaluated on both.

---

[3]Note that in the psychology literature, "theory of mind" typically refers to the attribution of mental states, such as phenomenological or epistemic states (i.e., perceptions or beliefs) to other intentional agents (Premack and Woodruff, 1978). In this paper, we address on only one potential component of theory of mind, present from early infancy, which focuses on reasoning about the intentional states, not the phenomenological or epistemic states, of others (Spelke, 2016).

## 2 Baby Intuitions Benchmark (BIB)

BIB focuses on the following questions: 1) Can an AI system represent an agent as having a preferred goal object? 2) Can it bind specific preferences for goal objects to specific agents? 3) Can it understand that physical obstacles might restrict agents' actions, and does it predict that an agent might approach a nonpreferred object when their preferred one is inaccessible? 4) Can it represent an agent's sequence of actions as instrumental, directed towards a higher-order goal object? 5) Can it infer that a rational agent will move efficiently towards a goal object?

Following the VOE paradigm, each of BIB's tasks includes a familiarization phase and a test phase, together referred to as an "episode." The familiarization phase presents eight successive trials introducing the main elements of the visual displays used in the test phase and allows the observer to form expectations about the future behavior of those elements based on their prior knowledge or learning. The test phase includes an unexpected and expected outcome based on what was observed during familiarization. Typically, the unexpected outcome is perceptually similar to the familiarization trials but is conceptually implausible, while the expected outcome is more perceptually different but involves no conceptual violation. The unexpected outcome is thus unexpected only if the observer possesses an abstract understanding of the events, and the expected outcome reflects a lower-level, heuristically, or perceptually based alternative. When VOE is used with infants, their looking time to each outcome is measured, and infants tend to look longer at the unexpected outcome (Baillargeon et al., 1985; Turk-Browne et al., 2008; Oakes, 2010).

Inspired by Heider and Simmel (1944), the primary set of visual stimuli present a fully observable "grid world," shown from an overhead perspective, and populated with simple geometric shapes that take on different roles (e.g. "agents," "objects," "tools") and provide few cues to those roles. We chose this type of environment as particularly suitable for testing AI systems (e.g., Baker et al., 2017; Rabinowitz et al., 2018) because it allows for procedural generation of a large number of episodes, and the simple visuals focus the problem on reasoning about agents. This design will also allow infancy researchers to test new questions about infant's understanding of agents in future work. While BIB and the baseline models tested here focus on this 2D grid-world environment, we have also instantiated the stimuli in 3D as a means of varying perceptual difficulty in future studies evaluating other models (appendix A).

### 2.1 Can an AI system represent an agent as having a preferred goal object?

**Developmental Background.** Infants infer that agents have preferences for goal objects, not goal locations (Gergely et al., 1995; Luo, 2011; Song et al., 2005; Woodward, 1998, 1999; Woodward and Sommerville, 2000). As illustrated in Figure 2 (left), Woodward (1998)'s seminal study showed that when 5- and 9-month-old infants saw a hand repeatedly reaching to a ball on the left over a bear on the right, they then looked longer when the hand reached to the left for the bear, even though the direction of the reach was more similar in that event to the events in the previous trials. These results suggest that the infants expected that the hand would reach consistently to a preferred goal object as opposed to a preferred goal location. Other studies have shown that infants' interpretations are not restricted to reaching events. For example, infants attribute a preference for goal objects to a 3D box during a live puppet show when that box seemingly exhibits self-propelled motion. (Luo, 2011; Luo and Baillargeon, 2005; Shimizu and Johnson, 2004). When shown an agent repeatedly moving to the same object at approximately the same location, do machines infer that the agent's goal is a preferred object, not location?

**Familiarization Trials.** The familiarization shows an agent repeatedly moving towards a specific object in a world with two objects (Figure 2a center). The agent's starting position is fixed, and the locations of the objects are correlated with their identities such that the preferred object and nonpreferred object appear in generally the same location across trials (appendix Figure 9 and 10).

**Test Trials.** The test uses two object locations that had been used during one familiarization trial, but the identity of the objects at those locations has been switched. In the expected outcome (Figure 2b), the agent moves to the object that had been their goal during the familiarization, i.e., their preferred object, but the trajectory of their motion and the location of that object is different from familiarization. In the unexpected outcome (Figure 2c), the agent moves to the nonpreferred object, but the trajectory of their motion and the location they move to is the same as familiarization. The model is successful if it expects the agent to go to the preferred object in a different location.

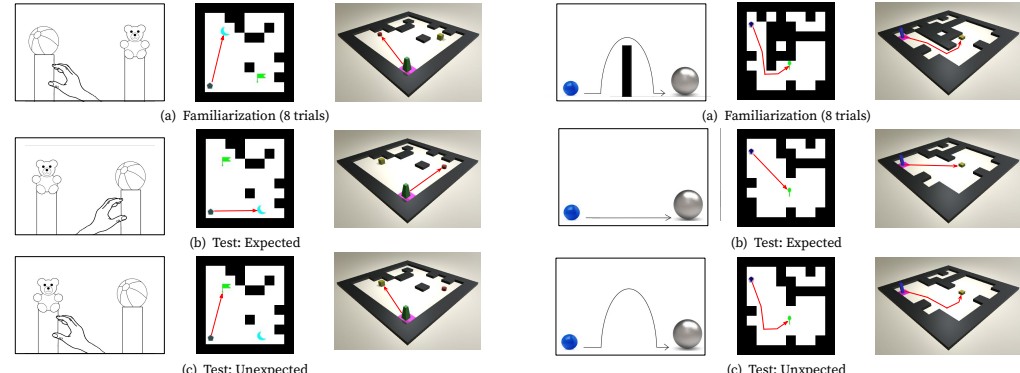

Figure 2: Can machines represent an agent's preferred goal object? Inspired by Woodward (1998)'s study with infants (left), BIB presents an agent navigating to their preferred goal object in approximately the same location across eight familiarization trials (a). At test, the location of the preferred goal object changes. The expected outcome (b) presents the agent moving to their preferred goal object in a new location, and the unexpected outcome (c) presents the agent moving to their nonpreferred object in the preferred object's old location. This evaluation has been rendered in 2D (middle) and 3D (right).

Figure 3: Can machines infer that rational agents move efficiently towards their goals? Inspired by Gergely et al. (1995) (left), BIB presents a rational agent navigating around an obstacle to its goal object across eight familiarization trials (a). At test, the rational agent either follows an efficient path (b) or an inefficient path (c).

## 2.2 Can an AI system bind specific preferences for goal objects to specific agents?

**Developmental Background.** Infants are capable of attributing specific object preferences to specific agents (Repacholi and Gopnik, 1997; Kuhlmeier et al., 2003; Buresh and Woodward, 2007; Henderson and Woodward, 2012). For example, while 9- and 13-month-old infants looked longer at test when an actor reached for a toy that they did not prefer during a habituation phase, infants showed no expectations when the habituation and test trials featured different actors (Buresh and Woodward, 2007). When shown one agent repeatedly moving to the same object, do machines expect that that object is preferred by that specific agent?

**Familiarization Trials.** The familiarization shows an agent consistently choosing one object over the other, but objects appear at widely varying locations in the grid world.

**Test Trials.** The test includes four possible outcomes: the same agent moves to the preferred object (expected); a new agent moves to the object preferred by the first agent (no expectation) (appendix, Figure 12); the same agent moves to the nonpreferred object (unexpected); or the new agent moves to the nonpreferred object (no expectation) (appendix, Figure 13). The model is successful if it has the same relative expectations as listed above; that is, weak or no expectations about the preferences of the new agent compared to the familiar agent.

## 2.3 Can an AI system understand that physical obstacles might restrict agents' actions, and does it predict that an agent might approach a nonpreferred object when the preferred one is inaccessible?

**Developmental Background.** Infants understand the principle of solidity (e.g., that solid objects cannot pass through one another), and they apply this principle to both inanimate entities (Baillargeon, 1987; Baillargeon et al., 1992; Spelke et al., 1992) and animate entities, such as human hands (Saxe et al., 2006; Luo et al., 2009). Infants' expectations about the objects agents might approach are also informed by object accessibility. Scott and Baillargeon (2013) demonstrate, for example, that 16-month-old infants expected an agent, facing two identical objects, to reach for the one in the container without a lid versus the one in the container with a lid. When shown an agent repeatedly moving to the same object, do machines recognize that the agent's access to that object might change, and do they predict that an agent might then approach a nonpreferred object?

**Familiarization Trials.** The familiarization shows an agent consistently choosing one object over the other, as above, and objects appear at widely varying locations in the grid world (Figure 4).

**Test Trials.** The test presents two new object locations. In the no-expectation outcome, the preferred object is now blocked on all sides by fixed, black barriers, and the agent moves to the nonpreferred object. In the unexpected outcome, both of the objects remain accessible, and the agent moves to the nonpreferred object (Figure 4). The model is successful if it has the same relative expectations; that is, weak or no expectation about the agent's moving to the nonpreferred object only when the preferred object is inaccessible.

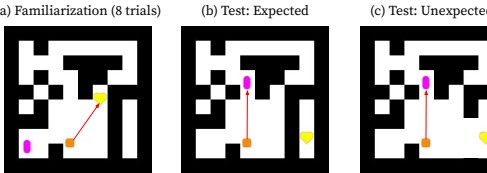

(a) Familiarization (8 trials)  (b) Test: Expected  (c) Test: Unexpected

Figure 4: Can machines understand that obstacles restrict actions? The familiarization trials present an agent navigating to their preferred object in varied locations (a). At test, either the preferred object is inaccessible and the agent goes to the nonpreferred object (b) or the preferred object is accessible and the agent goes to the nonpreferred object (c).

### 2.4  Can an AI system represent an agent's sequence of actions as instrumental, directed towards a higher-order goal object?

**Developmental Background.** Infants represent an agent's sequence of actions as instrumental to achieving a higher-order goal (Carpenter et al., 2005; Elsner et al., 2007; Hernik and Csibra, 2015; Gerson et al., 2015; Saxe et al., 2007; Sommerville and Woodward, 2005; Woodward and Sommerville, 2000). For example, Sommerville and Woodward (2005) showed that 12-month-old infants understand an actor's pulling a cloth as a means to getting the otherwise out-of-reach object placed on it. When shown an agent repeatedly taking the same action to effect a change in the environment that enables them to move towards an object, do machines expect that that object is the goal, as opposed to the sequence of actions?

**Familiarization Trials.** The familiarization includes five main elements: an agent; a goal object; a key; a lock; and a green removable barrier (see Figure 5). The green barrier initially restricts the agent's access to the object. And so, the agent removes the barrier by collecting and inserting the key into the lock. The agent then moves to the object.

**Test Trials.** The test phase presents three different scenarios for a total of six different outcomes. In the scenario with no green barrier: the agent moves directly to the object (expected); or to the key (unexpected) (Figure 5a). In the scenario with an inconsequential green barrier: the agent moves directly to the object (expected); or to the key (unexpected) (Figure 5b). In the scenario with variability in the presence/absence of the green barrier: the barrier blocks the agent's access to the object, and the agent moves to the key (expected); or, the barrier does not block the object and the agent moves to the key (unexpected). The model is successful if it expects the agent to go to the key only when the green removable barrier is blocking the goal object (Figure 5c).

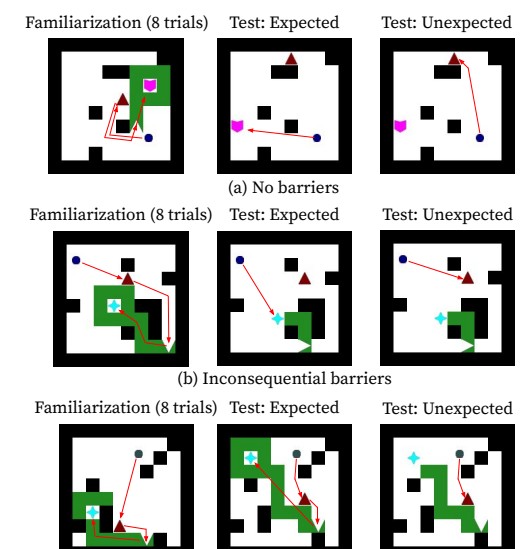

Familiarization (8 trials)  Test: Expected  Test: Unexpected

(a) No barriers

Familiarization (8 trials)  Test: Expected  Test: Unexpected

(b) Inconsequential barriers

Familiarization (8 trials)  Test: Expected  Test: Unexpected

(c) Blocking barriers

Figure 5: Can machines recognize instrumental actions towards higher-order goals? BIB's three types of test trials evaluate machines' understanding of instrumental actions. The agent's goal is initially inaccessible (blocked by a green removable barrier). During familiarization (left), the agent removes the barrier by retrieving the key (triangle) and inserting it into the lock. At test, the agent's moving directly to the goal is expected when the green barrier is absent (a) or not blocking the goal object (b,c); its moving to the key in those cases is unexpected.

### 2.5 Can an AI system predict that a rational agent will move efficiently towards a goal object?

**Developmental Background.** Infants expect agents to move efficiently towards their goals (Gergely et al., 1995; Gergely and Csibra, 1997, 2003; Liu et al., 2017, 2019; Colomer et al., 2020). In a seminal study by Gergely et al. (1995), for example, 12-month-old infants repeatedly saw a small circle jumping over an obstacle to get to a big circle (see Figure 3 left). At test, the obstacle was removed, and the small circle either took the same, now inefficient, path to get to the big circle or took the straight, efficient path. Infants were surprised when the agent took the familiar, inefficient path. These findings have been replicated by instantiating the agent and object in different ways (as, e.g., humans, geometric shapes, or puppets) and by using different kinds of presentations (e.g., prerecorded or live) (Colomer et al., 2020; Phillips and Wellman, 2005; Sodian et al., 2004; Southgate et al., 2008; Liu et al., 2017). When infants see an irrational agent, i.e., one moving inefficiently to their goal from the start, however, they do not form any expectations about that agent's actions at test (Gergely et al., 1995; Liu and Spelke, 2017). When shown a rational agent repeatedly taking an efficient path around an obstacles to its goal object, do machines expect that that agent will continue to take efficient paths, as opposed to similar-looking paths, relative to the obstacles in the environment?

**Familiarization Trials** The familiarization includes two different scenarios: a rational agent consistently moves along an efficient path to its goal object around a fixed black barrier in the gird world (Figure 3a); or, an irrational agent moves along these same paths as the rational agent, but there is no barrier in the way (appendix, Figure 14b).

**Test Trials.** The test includes two possible scenarios. One scenario shows only the rational agent, and it presents one of the familiarization trials but with the barrier between the agent and the goal object removed or changed in position (such that a curved path is still required). The agent either moves along an efficient path to its goal (expected) or the agent moves along one of two unexpected paths, either the exact same, but now inefficient, path that it had during familiarization (path control, Figure 3) or along a path that is inefficient but takes the same amount of time as the efficient path (in this latter case, the goal object starts off closer to the agent, appendix Figure 11). The second scenario shows either the rational or irrational agent taking an inefficient path towards its goal. This outcome should be unexpected in the case of the rational agent, but should yield no expectation in the case of the irrational agent (appendix, Figure 14). The model is successful if it expects only a rational agent to modify its path based on the location of barriers and move efficiently to its goal.

## 3 Background Training

While infants in the lab make meaningful inferences about novel stimuli and environments with only a brief familiarization phase, BIB includes tens of thousands of background episodes as a generous stand-in for this type of in-lab familiarization. Models should therefore not be surprised merely by BIB's elements and dynamics. Just as infants may already have knowledge about agents, objects, and places prior to coming to the lab, moreover, we do not intend to limit models to just BIB's background training prior to being tested. Although learning-centric approaches will learn something about agents if trained on the background set, either supplemental pretraining or additional prior knowledge can be enriched by the background training for a model to approach the benchmark successfully. The episodes in the background training are structured similarly to those in the evaluation, although the familiarization and test trials are drawn from the same distribution for the background training only. Similar to IntPhys (Riochet et al., 2018) and ADEPT (Smith et al.,

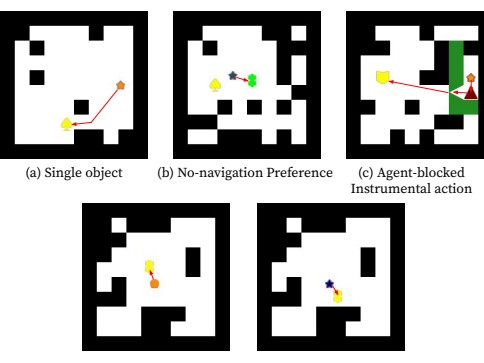

(a) Single object  (b) No-navigation Preference  (c) Agent-blocked Instrumental action

(d) Single-object Multiple-agents

Figure 6: The four tasks from the background training set, including the Single-Object Task (a), the No-Navigation Preference Task (b), the Agent-Blocked Instrumental Action Task (c), and the Single-Object Multiple-Agent Task (d). For tasks (a)-(c), only the test trials are shown here. For task (d), the agent switches during familiarization and continues during test. An example of two agents in the same episode are shown here.

2019), we only provide expected outcomes at test for the background training. There are four training tasks:

**Single-Object Task.** The agent navigates to an object at some varied location in the environment (Figure 6a). This task is different from the evaluation tasks, where there are two objects or the arrangement of the barriers in the environment changes. With this training, models can learn how agents start and end trials, how agents move, and how barriers influence agents' motion. We provide 10,000 episodes of this type.

**No-Navigation Preference Task.** Two objects are located very close to the agent's initial starting location but at varied locations, and the agent approaches one object consistently across trials (Figure 6b). The task allows the model to learn that agents display consistent preference-based behavior. Critically, the navigation in these trials is trivial compared to the evaluation trials, so navigation to a preferred goal object is not trained. We provide 10,000 episodes of this type.

**Single-Object Multiple-Agent Task.** One object is located very close to the agent's initial starting location but at varied locations (Figure 6d). At some point during the episode, a new agent takes the initial agent's place (for example, the initial agent could be replaced at the fourth trial, and all subsequent trials, including the test trial, would have the new agent). The task allows the model to learn that multiple agents can appear across trials. This task differs from the evaluation task, where there are two objects and a new agent appears only in the test trials. We provide 4,000 episodes of this type.

**Agent-Blocked Instrumental Action Task.** The trial starts with the agent confined to a small region of the grid world, blocked by a removable green barrier (Figure 6c). The agent collects the key and inserts it into the lock to make the barrier disappear. The agent then navigates to the object. This task allows the model to learn that the green barrier obstructs navigation and that inserting the key in the lock removes that barrier. These trials differ from the evaluation in that the removable barriers are around the agent instead of the object. We provide 4,000 episodes of this type.

To be successful at the evaluations, models must acquire or enrich their representations of agents for flexible and systematic generalization. For example, models have to combine acquired knowledge of navigation (Single-Object Task) and preferences (No-Navigation, Preference Task) to be successful at the evaluation testing their understanding that agents have preferences for goal objects, not goal locations (section 2.1).

## 4    Baseline Models

When being evaluated on BIB, a model cannot actively sample from the environment; it can only use the samples provided in the episodes themselves. We therefore did not test baseline models using traditional approaches in imitation learning (IL), inverse RL (IRL), and RL (Ng et al., 2000; Abbeel and Ng, 2004; Ziebart et al., 2008) because they require substantial privileged information, such as access to the environment to actively sample trajectories using the modelled policy, and, in the case of the RL algorithms, an observable reward. Moreover, these approaches often model one agent at a time, and BIB requires the same model to infer the behavior of different agents across different episodes (although recent approaches in deep RL and IRL try to mitigate this latter issue with work in meta-RL and meta-IRL (Xu et al., 2019; Yu et al., 2019; Rakelly et al., 2019) that allows for similar cross-episode adaptation). Although this feature of BIB makes it less suitable for testing RL models, it is essential to BIB's design because it reflects infants' reasoning. Infants rely on little to no active interaction with a particular environment to make meaningful inferences and predictions about the agents in that environment, and infants' inferences are far more abstract than their particular observed or active experience (Skerry et al., 2013; Gergely et al., 2002; Liu et al., 2019; Zmyj et al., 2009).

We thus tested three baseline models spanning different approaches including video modeling, behavior cloning (BC), and offline RL (see appendix C for full model specifications). Models were trained passively and through observation only. An episode in BIB can be broken down into nine trials with trajectories, $\{\tau_i\}_{i=1}^9$, where $\tau_i \forall i \in [1, 8]$ are the familiarization trials and $\tau_9$ is the test trial. Each trajectory $\tau_i$ consists of a series of state (frames from the video) and action pairs $(s_{ij}, a_{ij})_{j=1}^T$. The action space of the agents in BIB is of size eight; agents can move along the two axes or along the diagonals. Additionally, the transitions are deterministic.

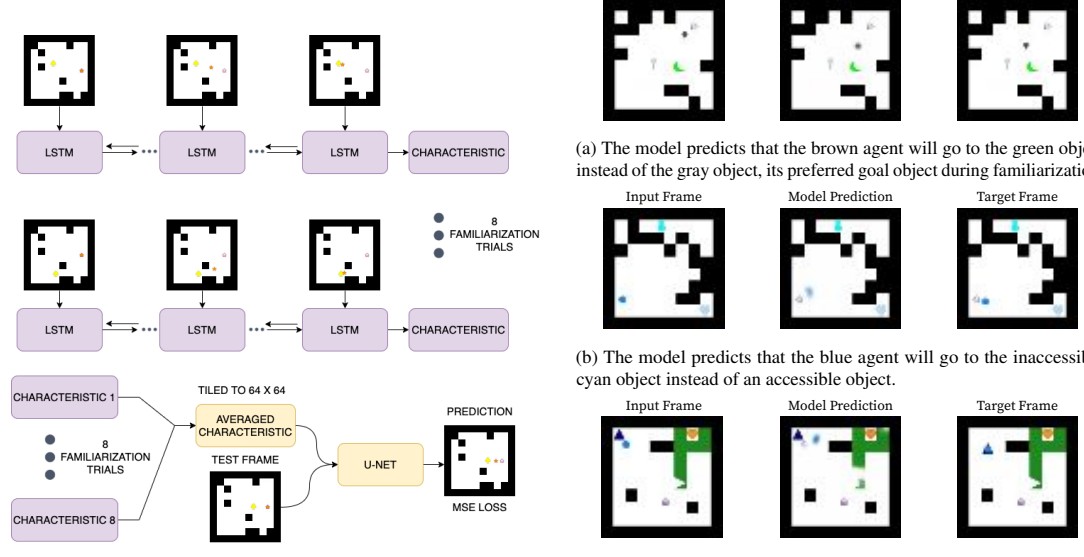

(a) The model predicts that the brown agent will go to the green object instead of the gray object, its preferred goal object during familiarization.

(b) The model predicts that the blue agent will go to the inaccessible cyan object instead of an accessible object.

(c) The model predicts that the blue agent will go to the inaccessible orange object instead of performing the instrumental action to first collect the triangular key.

Figure 7: Architecture of the video baseline model inspired by Rabinowitz et al. (2018). An agent-characteristic embedding is inferred from the familiarization trials using a recurrent net. This embedding, with a frame from the test trial, is used to predict the next frame of the video using a U-Net (Ronneberger et al., 2015).

Figure 8: The most surprising frame (the frame with the highest prediction error) from the test trial for the video model taken from the evaluation tasks. Examples of failed expectations are shown here.

Our baseline models either predict the next frame in the video (see Figure 7 for architecture) or the actions taken by the agent (see appendix C.2). To encode the context in the form of the familiarization trials, we use a sequence of frames (for the video model) and frame-action pairs (for the BC models). In terms of the architecture, the baseline models take inspiration from a state-of-the-art, neural-network-based approach to encode the characteristic of an agent: the theory of mind net (ToMnet) model in Rabinowitz et al. (2018). We encode the familiarization trials as context using either a bidirectional LSTM or an MLP. In addition to video modeling and BC, we also try an offline-RL baseline (Siegel et al., 2020). Offline-RL algorithms (Levine et al., 2020) are designed to learn a policy from demonstrations by using privileged information in the form of rewards received by the agent. We engineer a reward function based on the distance of the agent from the goal to train the RL policy (see appendix C.3).

During evaluation, the "expectedness" of a test trial, in the context of the previous familiarization trials, was inferred by a model's error on the most 'unexpected' step (i.e., the step with the highest prediction error). The most 'unexpected' step was chosen for comparison as alternatives (like the mean expectedness of steps) consistently resulted in lower VOE scores. For each evaluation episode, we first calculated the model's relative accuracy, i.e., whether the model found the expected video in each pair more expected than the unexpected video (chance is 50%).

## 5   Results

The models were trained on 80% of the background training episodes (training set), and the rest of the episodes were used for validation (validation set). A comparison of the MSE loss (on pixels for the video model and in the action space for the BC and RL models) on the training and validation sets indicated that the models had learned the training tasks successfully (see appendix C).

The results of our baselines are presented in Table 1. The offline RL model performs similarly to the BC approach (see appendix C.3 for details), so we do not consider the offline RL model further (offline RL is known to provide little improvement over BC when the demonstrations are not noisy (Siegel et al., 2020)). Comparing the performance of BC on BIB and AGENT (Shu et al.,

| BIB AGENCY TASK | BC-MLP | BC-RNN | VIDEO-RNN |
|---|---|---|---|
| PREFERENCE | 26.3 | 48.3 | 47.6 |
| MULTI-AGENT | 48.7 | 48.2 | 50.3 |
| INACCESSIBLE GOAL | 76.9 | 81.6 | 74 |
| EFFICIENCY: PATH CONTROL | 94.0 | 92.8 | 99.2 |
| EFFICIENCY: TIME CONTROL | 99.1 | 99.1 | 99.9 |
| EFFICIENCY: IRRATIONAL AGENT | 73.8 | 56.5 | 50.1 |
| EFFICIENT ACTION AVERAGE | 88.8 | 82.5 | 83.1 |
| INSTRUMENTAL: NO BARRIER | 98.8 | 98.8 | 99.7 |
| INSTRUMENTAL: INCONSEQUENTIAL BARRIER | 55.2 | 78.2 | 77.0 |
| INSTRUMENTAL: BLOCKING BARRIER | 47.1 | 56.8 | 62.9 |
| INSTRUMENTAL ACTION AVERAGE | 67.0 | 77.9 | 79.9 |

Table 1: Performance of the baseline models on BIB. The scores quantify pairwise VOE judgements.

2021), we see that BC performs worse on BIB. BC and offline-RL are known to perform poorly on out-of-distribution and systematically different scenarios from training to test (Siegel et al., 2020). Because systematic generalization is a primary feature of BIB but not AGENT, this is likely why BC performs worse on BIB.

The two models we tested with an RNN (Video-RNN; BC-RNN) perform at chance on the Preference Task (see Figure 8a for predictions made by the video model); they tend to predict that an agent will go to the closer object (this prediction is made in about 70% of trials). The models thus ignore the agent's preference, established during familiarization. This finding is especially striking relative to the models' success on the No-Navigation, Preference Task from the background training and could result from differences in the distance at which the objects are placed in the scene. In the background training, the objects are close to the agent, yielding short trial lengths during familiarization. The characteristic encoder RNN might find it difficult to generalize to the evaluation tasks' longer sequences. The BC-MLP model is confused by how the object locations correlate with their identity, encoding an agent's preference for a goal location instead of a goal object. This too is surprising, as the background training provides evidence that agents prefer object identities, not specific locations. None of these models, as a result, succeed in recognizing that agents have preferences and goals for specific objects.

The models also fail on the Multi-Agent Task, again tending to predict that an agent will go to the closer object regardless of any established preferences. Consistent with this failure, the models also fail to map specific preferences to specific agents. The models do slightly better than chance on the Inaccessible Goal Task. As seen in Figure 8b, the video model still, nevertheless, frequently predicts that the agent will go to the inaccessible goal. The models are proficient at finding the shortest path to the goal in the Efficiency Task (appendix Figure 16a), leading to high accuracy on both sub-evaluations that test for efficient action: Path Control and Time Control (Table 1). However, the two models with an RNN fail to modulate their predictions based on whether the agent was rational or irrational during familiarization (Table 1). The BC-MLP has a weak expectation of rationality from an irrational agent, scoring slightly higher that the two RNN models on this task. Finally, the models perform above chance on the Instrumental Action Task, but performance on the sub-evaluations (Table 1) indicate that they rely on the simple heuristic of directly going to the goal object rather than understanding the nature of the instrumental action (Figure 8c). This heuristic leads to higher scores on sub-evaluations with no barrier and an inconsequential barrier (Table 1) but lower scores on the sub-evaluation with a blocking barrier. This poor performance may be due to the difference in the relations between the agent and barrier in the background training (where the agent is confined; Figure 6c) and evaluation (where the object is confined; Figure 5).

## 6  General Discussion

We introduced the Baby Intuitions Benchmark (BIB), which tests machines on their ability to reason about the underlying intentionality of other agents by only observing their actions. BIB is directly inspired by the abstract reasoning about agents that emerges early in human development, as revealed by behavioral studies with infants. BIB's adoption of the VOE paradigm, moreover, means both that its results can be interpreted in terms of human performance and that low-level heuristics can be

directly evaluated. While baseline, deep-learning models successfully generalize to BIB's training tasks, they fail to systematically generalize to the evaluation tasks even though the models incorporate theory-of-mind-inspired architectures (Rabinowitz et al., 2018).

What kinds of models might succeed on BIB? Shu et al. (2021) proposes a Bayesian inverse planning model for success on AGENT's agency-reasoning tasks. Extending this structured Bayesian approach to BIB is not straightforward, however, as it requires features that are not currently provided. Recent approaches in deep imitation learning and inverse RL, especially work in meta-IL and meta-IRL (Xu et al., 2019; Yu et al., 2019) that allows for test-time adaptation, also show promise and could ultimately lead to more human-like reasoning about other agents. Extending these approaches to BIB is nevertheless also non-trivial since they require active sampling of the environment, which is not something that BIB allows. We hope that BIB will catalyze future research in these directions.

While the comparison between human and machine performance will be bolstered by future behavioral studies with infants, the results from the baseline models already shed new light on critical differences between human and artificial intelligence (also see appendix D). For example, across tasks, the baseline models tended to predict that an agent would move to the closer of two objects, regardless of the agent's previously demonstrated preference for only one of the objects. And, while models succeeded in predicting that an agent would move efficiently to an object, they over-generalized that efficiency principle to include agents who had previously demonstrated inefficient, irrational actions. For infants, object-based preferences and efficient, goal-directed action instead serve to enrich their understanding of the intentions of others. For example, when infants observe an agent going farther to a particular object or exerting more effort to reach it, they can attribute both a preference for that object to that agent and a value to that preference (Liu et al., 2017).

BIB also raises new questions about the foundations of common-sense reasoning about agents for both humans and machines alike. For example, the "extended familiarization" needed for training AI models (i.e., the background training) potentially reveals a striking difference between how BIB might challenge minds versus machines. While both infants and AI systems may have built-in knowledge and/or experience prior to participating in BIB, infants likely need only eight, as opposed to thousands, of videos of shapes moving around grid worlds to successfully apply their reasoning about agents to new, test events presented in that medium. Nevertheless, it remains unknown whether infants might also benefit from some kind of background training meant for machines; such designs have never been tested with infants. Relatedly, BIB challenges the generalizability of early emerging, human common-sense reasoning about agents. How well do humans recognize simple shapes with simple movements and minimal cues to animacy (e.g., no eyes/gaze direction, no distinctive sounds, and no emotional expressions) as agents with intentionality? Do navigation- versus reaching-contexts deferentially shift humans' attention to locations versus objects as candidates for an agent's goal? How does a comprehensive set of agency-reasoning abilities relate to one another within the same individual? Most of the existing infant literature on which BIB is based presents infants with richer cues to animacy (in the form of live-action or animated displays from frontal or three-quarters points of view), events with no or minimal navigation to establish preferences for goal objects, and individual tests of one competency or another, outside of a unified framework.

The origins and development of humans' intuitive understanding of agents and their intentional actions have been studied extensively in developmental cognitive science. The representations and computations underlying such understanding, however, are not yet understood. BIB serves as a test for computational models with different priors and learning-based approaches to achieve common-sense reasoning about agents like human infants. A computational description of how we reason about agents could ultimately help us build machines that better understand us and that we better understand.

Finally, BIB serves as a key step in bridging machines' impoverished understanding of others' mental states with humans' rich one. To achieve a human-like theory of mind, a model would not only have to understand intentionality, which is tested in BIB, it would also have to understand the perceptions and beliefs of other agents. A combined, comprehensive understanding may underlie human theory of mind and lead to, for example, success in a false belief task, in which humans have expectations about where an agent would search for a goal object based on where that agent last saw it (Baron-Cohen et al., 1985; Spelke, 2016). A benchmark that focuses on reasoning about agents' phenomenological and epistemic states is thus a natural extension of BIB and could further advance our understanding of both human and artificial intelligence.

## Acknowledgments and Disclosure of Funding

This worked was supported by the DARPA Machine Common Sense program (HR001119S0005). We thank Victoria Romero, Koleen McKrink, David Moore, Lisa Oakes, and Clark Dorman for their generous feedback. We are also grateful to Thomas Schellenberg, Dean Wetherby, and Brian Pippin for their development effort in porting the benchmark to 3D. Finally, we thank Brian Reilly for coming up with the name of the benchmark and finding the perfect acronym for our work.

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
