

(b) Test: Expected

(a) Familiarization Trials

(c) Test: Unexpected

Figure 9: Evaluation to test if machines can represent an agent's preference for a goal object not a goal location. 2D versions of the stimuli are shown here.

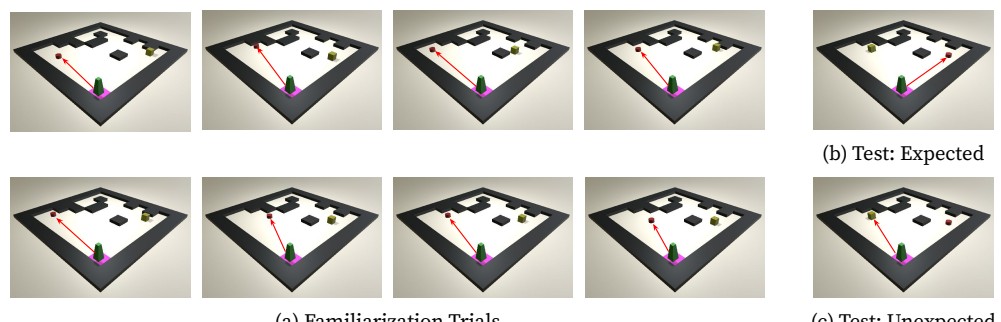

(b) Test: Expected

(a) Familiarization Trials

(c) Test: Unexpected

Figure 10: Evaluation to test if machines can represent an agent's preference for a goal object not a goal location. 3D versions of the stimuli are shown here.

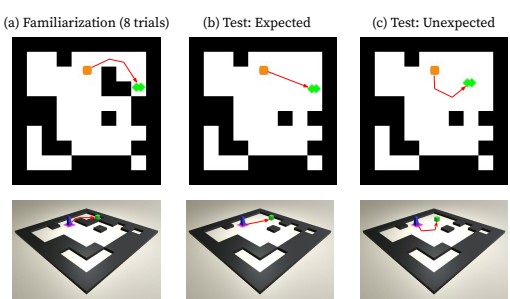

(a) Familiarization (8 trials)  (b) Test: Expected  (c) Test: Unexpected

Figure 11: We draw inspiration from Gergely et al. (1995) to ask whether machines can understand that rational agents move efficiently towards their goals. In this task, the time taken by the agent to reach the goal in the expected and unexpected outcomes is the same (time control).

# A   Generating the Evaluations

For each of the five evaluation tasks, we generated 1000 episodes, each with one expected and one unexpected outcome (2000 videos), by sampling the locations of barriers, agents, and objects in the $10 \times 10$ grid. The locations were controlled to account for the distances and obstacles between the agent and the objects so that, e.g., preferred objects were not consistently closer or farther from agents. We provide two evaluation sets, one with objects and agents seen during the background training and the other with new objects and agents. Finally, as a means of varying the perceptual difficulty of

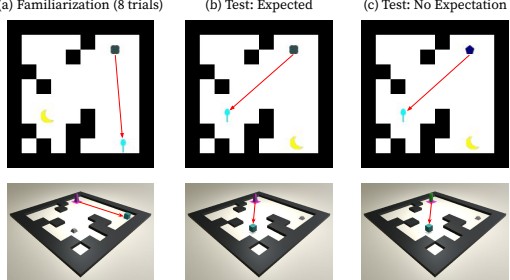

Figure 12: Evaluation to test if machines can bind specific preferences to specific agents. The gray agent consistently chooses the cyan object over the yellow object (a) . The same gray agent moves to the preferred cyan object (b). The new agent moves to the preferred (by the first agent) cyan object (c)

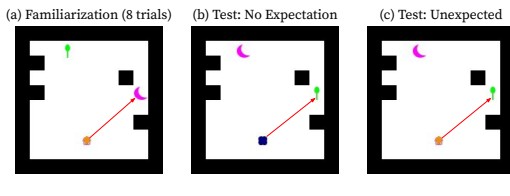

.

Figure 13: Evaluation to test if machines can bind specific preferences to specific agents. The orange agent consistently chooses the pink object over the green object (a) . A new blue agent moves to the nonpreferred (by the first agent) green object (b). The same orange agent moves to the nonpreferred green object (c)

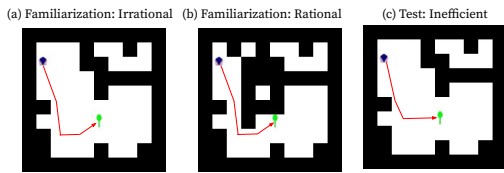

Figure 14: Inspired by Gergely et al. (1995), we ask whether machines can differentiate between rational or irrational agents in terms of their efficient action.

the benchmark, we also include 3D versions of the stimuli rendered to match the 2D versions and presented at a three-quarters point of view (Figure 2).The 2D stimuli (except for the instrumental action tasks) are directly translated to 3D using the AI2THOR (Kolve et al., 2019) framework. For both 2D and 3D videos, we provide scene configuration files describing the objects and agents present in the scene.

## B   Data Specifications

Each video has a resolution of 200 x 200 at 25 fps (the videos can be converted to a higher resolution if required). In addition to the videos, we provide metadata in the form of json files describing every frame in the video. This description contains information about the layout of the scene and the elements present.

Each video has a json file associated with it. A video has nine trials which correspond to the nine items in the json file. These nine trials have a variable number of frames. Each frame is described by the elements contained in it.

These include:

- The 'size' attribute specifies the resolution of the frame.

- The 'walls' attribute has a list of [bottom-left, extent] attributes describing the barriers. The bottom-left attribute is 2-dimensional and is defined by an x and y coordinate. Similarly, the extent for each wall is 2-dimensional and describes the width and height of the wall.

- The 'objects' attribute is defined as a list of attributes [bottom-left, size, image, color]. The bottom-left attribute is 2 dimensional and is defined by an x and y coordinate. The size is the half of the side of the square shape that the image of the object would be resized to. So, if the size is 10, an object image of size 100x100 would be resized to 20x20. The image attribute gives the path of the object image. The color attribute gives the color of the object in RGB format in the range [0, 255].

- The 'home', 'agents', 'key' and 'lock' attributes have a similar structure to the objects attribute.

- The 'fuse' attribute corresponds to the removable barrier and has a similar structure to the 'walls' attribute.

## C    Baseline Details

### C.1    Video Model

**Model Description.** The models (see Figure 7) operate on videos sampled at 3 fps and resized to $64 \times 64$. Each frame in each familiarization trial is encoded using a convolutional neural network. The frame embeddings in a trial are passed to a bidirectional LSTM. The last output embdedding of the LSTM represents the characteristic of the agent in the trial. These embeddings are averaged across familiarization to obtain a characteristic embedding for an agent. The characteristic embedding is tiled to a $64 \times 64$ spatial resolution, concatenated to a frame from the test trial, and passed through a U-net to predict the next frame in the trial. A mean squared error loss is used to train the network.

In the video model, the frames of a familiarization trial are encoded using a residual convolutional network with four blocks, each with two $3 \times 3$ convolutional operations with 16 feature maps. This is followed by a $1 \times 1$ convolutional layer to map the 16 feature maps to one map. This representation is flattened and passed sequentially to a bi-directional LSTM. The output from the last timestep is used as the agent characteristic representation of size $1 \times 16$ for the trial (see Figure 7). The characteristic embedding across the eight familiarization trials is averaged to get a final agent characteristic embedding. This embedding is tiled to get a vector of size $64 \times 64 \times 16$ and concatenated to the current frame from the test trial. This vector of size $64 \times 64 \times 19$ is passed to a U-Net (Ronneberger et al., 2015) to predict the next frame. We use an MSE loss in the pixel space to train the model and an Adam optimizer with a learning rate of 1e-4 (betas=(0.9, 0.999)). We train the 2D video model for 11 epochs and the 3D model for ten epochs.

**Background Training.** The errors on the validation set for the model are shown in appendix Table 2. Some of the predictions made by the model can be seen in Figure 15. Only the Preference Task requires the model to take the familiarization phase into consideration. **Evaluation Tasks.** The model

| BIB TASK | MSE |
|---|---|
| SINGLE OBJECT | $3.3 \times 10^{-4}$ |
| PREFERENCE | $5.4 \times 10^{-4}$ |
| MULTI-AGENT | $2.4 \times 10^{-4}$ |
| INSTRUMENTAL ACTION | $9 \times 10^{-4}$ |

Table 2: The performance of the video model on the 2D background training tasks.

fails to reliably understand the agent's preference. This could be a result of differences in the distance at which the objects are placed in the scene. In the background training, the objects are placed close (section 3) to the agent, making the length of the familiarization trials short. The characteristic encoder LSTM might find it difficult to extract characteristics from the longer sequences in the evaluation tasks.

The model learns the simple heuristic of always going to the object in the Instrumental Action Task. This result could be due to a difference in the distribution of the background training and evaluation tasks. In the background training (Figure 6c), the agent is confined to a small space, blocked by the green removable barrier. The number of samples that the model has to predict for the agent's movement to the key or the lock is relatively small compared to the number of samples for the barriers disappearing and the agent's moving towards the object goal. In the evaluation tasks (Figure 5c), the agent's movement to the key and the lock are significantly greater (as the object goal is now blocked by the removable barrier). The model thus has trouble generalizing to this case (Table 1 Instrumental: Blocking barriers task).

When we replace the elements in the evaluation set with new ones, moreover, the video model scores fall slightly, but the trends remain the same (Table 3). Finally, the video model performs similarly on the 3D videos of the tasks, although performance is generally worse overall with 3D videos. This is likely because perceiving the trajectories of agents in 3D is more difficult for a predictive model in pixel space. The predictive networks trained with MSE find it challenging to model trajectories in depth.

| BIB AGENCY TASK | NEW OBJECTS | 3D VIDEOS |
|---|---|---|
| PREFERENCE | 47.4 | 49.2 |
| MULTI-AGENT | 50.0 | 50.0 |
| INACCESSIBLE GOAL | 61.7 | 40.0 |
| EFFICIENCY: PATH CONTROL | 98.5 | 66.3 |
| EFFICIENCY: TIME CONTROL | 96.9 | 75.4 |
| EFFICIENCY: IRRATIONAL AGENT | 47.8 | 50.0 |
| EFFICIENT ACTION AVERAGE | 72.7 | 62.9 |
| INSTRUMENTAL: NO BARRIER | 93.0 | - |
| INSTRUMENTAL: INCONSEQUENTIAL BARRIER | 66.0 | - |
| INSTRUMENTAL: BLOCKING BARRIER | 59.7 | - |
| INSTRUMENTAL ACTION AVERAGE | 69.6 | - |

Table 3: Performance of the video model on BIB with new objects and on the 3D videos. The scores quantify pairwise VOE judgements.

## C.2  Behavior Cloning

In behavior cloning, we try to predict the actions of an agent given the current state and context (in the form of familiarization trials). We sample the videos (originally at 30fps) at 3fps and resize the frames (originally $200 \times 200$) to $84 \times 84$. We define the action that an agent takes from one state to another as the change in its location. This results in a 2-dimensional continuous action space where values are in the range from $[-20, 20]$. We normalize the values so that they are in the range $[-1, 1]$.

**Model Description.** There are three components to the behavior cloning model: the state encoder; the context encoder; and the policy network. The architecture is similar to that of Yu et al. (2019); Rakelly et al. (2019); Rabinowitz et al. (2018). First, the states (frames) are encoded with a CNN (state encoder). We pretrain the state encoder using Augmented Temporal Contrast (ATC) (Stooke et al., 2021), which uses a contrastive loss to predict future state embeddings with random shift augmentations. We train the encoder on videos from the background training set for 350K iterations using ADAM with a learning rate of $1e - 4$ and default parameters. See Table 4 for specifications.

The second part of the BC model is the context encoder. States from the familiarization trials are encoded using the context encoder and these embeddings are concatenated to the actions. We either use a bidirectional LSTM or an MLP to encode the context. When using a bidirectional LSTM, we sample a sequence from each familiarization trial (with a max length of 30) to extract a trial embedding. This is averaged across the eight trials to get the characteristic embedding (similar to the video model). When using an MLP, we randomly sample 30 state-action pairs (state embedding concatenated with action). Each of these pairs is encoded using an MLP to get a transition embedding. The 30 transition embeddings are averaged to get a characteristic embedding.

| Input Frame | Model Prediction | Target Frame |
| --- | --- | --- |

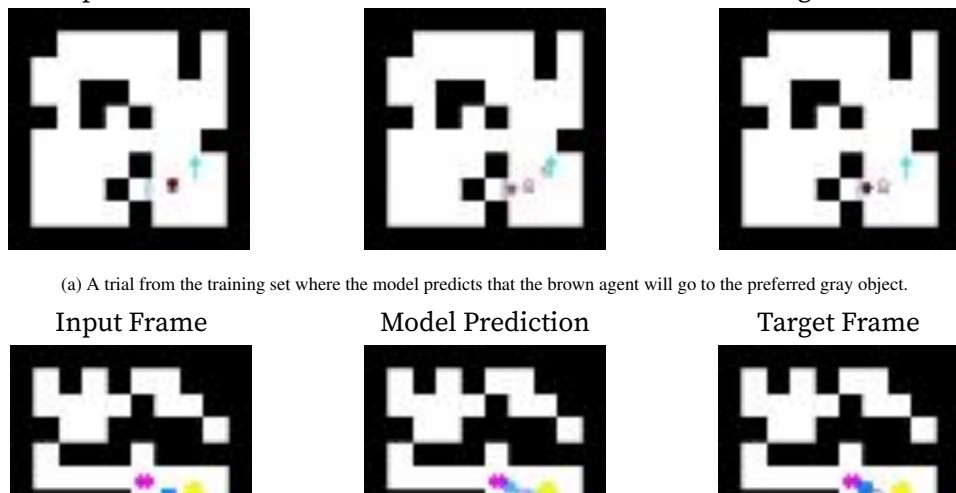

(a) A trial from the training set where the model predicts that the brown agent will go to the preferred gray object.

| Input Frame | Model Prediction | Target Frame |
| --- | --- | --- |

(b) A trial from the training set where the model predicts that the blue agent will go to the preferred magenta object . We see that there is blurred blue prediction close to the yellow object but the model thinks that it is more likely that the agent will go to the magenta one.

| Input Frame | Model Prediction | Target Frame |
| --- | --- | --- |

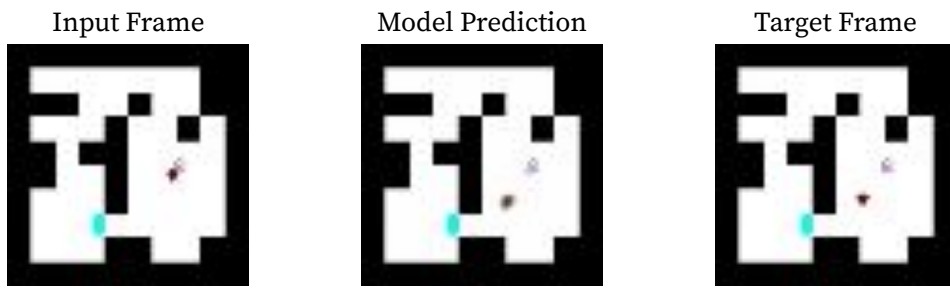

(c) The model correctly predicts that the agent will take the shortest path to go to the goal object.

| Input Frame | Model Prediction | Target Frame |
| --- | --- | --- |

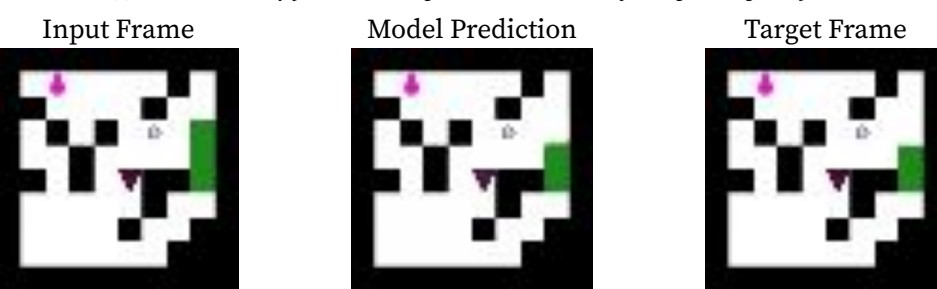

(d) The model correctly predicts that in the Instrumental Action Task, when the key is inserted into the lock, the removable barriers will slowly disappear.

Figure 15: Predictions of the video model on the background training tasks. (a) and (b) show model predictions for two preference trials where the model splits its predictions between the two objects but thinks that going to the preferred object is more likely. (c) shows model predictions for the Single Object Task where the model predicts that the agent will take the shortest path to the object. (d) shows the Instrumental Action Task where the model predicts the disappearance of the removable barriers. Test trials are shown here.

| Input Frame | Model Prediction | Target Frame |
| --- | --- | --- |

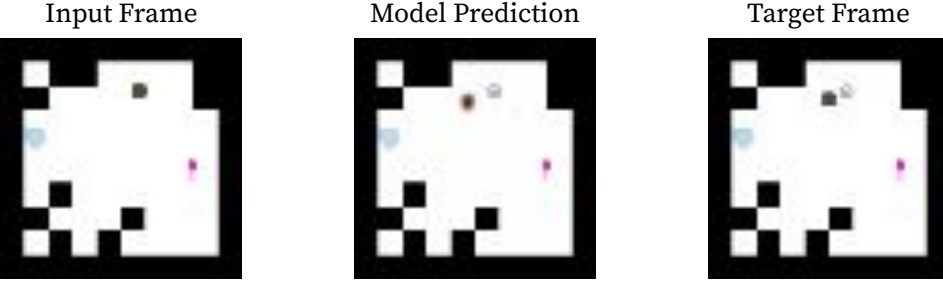

(a) Preference Task: The model correctly predicts that the brown agent will go to the preferred object (gray heart).

| Input Frame | Model Prediction | Target Frame |
| --- | --- | --- |

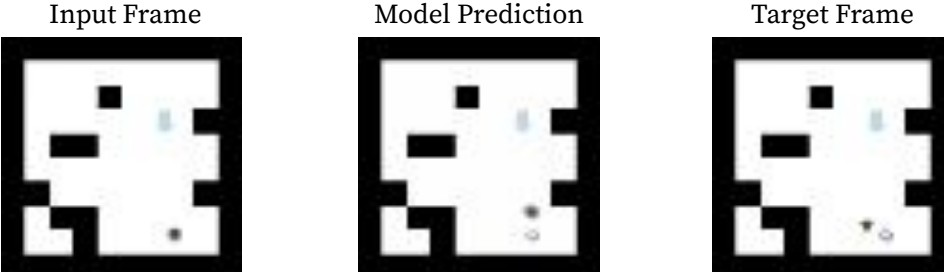

(b) Efficient Action task: The model correctly predicts that the brown agent will take the shortest path to its goal object. The target frame from the unexpected trial is shown here.

'

Figure 16: The most unexpected frame (the frame with the highest prediction error) from the test trial for the video model taken from the evaluation tasks. Successful examples are shown here.

| PARAMETER | VALUE |
| --- | --- |
| # OF CONV LAYERS | 4 |
| # OF FILTERS | 32 |
| KERNEL SIZE | $3 \times 3$ |
| EMBEDDING SIZE | 256 |
| TARGET UPDATE INTERVAL | 1 |
| TARGET UPDATE WEIGHT | 0.01 |
| RANDOM SHIFT PROBABLITY | 1 |

Table 4: Specifications of the state encoder model and parameters used to train it using augmented temporal contrast (ATC) (Stooke et al., 2021).

The final part of the BC model is the policy network. The current state (frame) is encoded using the state encoder. The state embedding is concatenated with the characteristic embedding. This state-context vector is passed through an MLP to predict the action of the agent. See Table 5 for specifications.

| PARAMETER | VALUE |
| --- | --- |
| CONTEXT LSTM LAYERS | 2 |
| CONTEXT LSTM HIDDEN SIZE | 32 |
| CONTEXT MLP HIDDEN SIZES | $[64, 64]$ |
| CONTEXT EMBEDDING SIZE | 32 |
| POLICY MLP HIDDEN SIZES | $[256, 128, 256]$ |

Table 5: Specifications of the BC model.

**Background Training.** We use an MSE loss in the action space to train the network. We train the models to convergence using ADAM with a learning rate of $5e - 4$. The model is successful on the background tasks (see Table 6).

| BIB TASK | BC-MLP | BC-RNN | OFFLINE RL |
|---|---|---|---|
| SINGLE OBJECT TASK | 0.05 | 0.05 | 0.09 |
| NO-NAVIGATION PREFERENCE TASK | 0.02 | 0.02 | 0.03 |
| NO-PREFERENCE MULTI-AGENT TASK | 0.05 | 0.03 | 0.05 |
| AGENT-BLOCKED INSTRUMENTAL ACTION TASK | 0.04 | 0.03 | 0.07 |

Table 6: Performance of the BC models and the Offline RL model on the validation set. MSE values of the predicted actions are shown above. An MSE value of 0.01 corresponds to the predicted action being off by two pixels, when the video frame is $200 \times 200$.

**Evaluation Tasks.** At test, we increase the number of samples passed as context to the BC-MLP model to 100. The expectedness of a transition is measured by the MSE error of predicting an action. The expectedness of an episode is formulated as the most 'unexpected' action in the test trial (we can use the average expectedness of the test trial as an alternate measure, but using the max was empirically found to perform better).

## C.3 Offline RL

There are several ways to learn from demonstrations using Offline RL/ Batch RL (Levine et al., 2020). We use a method defined in Siegel et al. (2020) that relies on constraining the policy using a policy prior. This policy prior is simply policy model that maximizes the likelihood of the observed data. Siegel et al. (2020) use either a simple Behavior Model (BM) (maximizes the likelihood of data similar to behavior cloning) an Advantage Weighted Behavior Model (ABM) (maximizes the likelihood of observed actions weighted by a learned advantage function). In addition to learning the prior, a Q-value estimator is learned to evaluate a policy. An RL policy is learned with the Q-value estimates and the behavioral prior using Maximum a posteriori policy optimization (Abdolmaleki et al., 2018). See Siegel et al. (2020) for details.

Siegel et al. (2020) show that when the demonstrations are not noisy and come from a reliable expert, the Q-network can be learned using the prior policy. The RL policy improvement step can be performed independently from the prior learning and Q-value estimation step. Since the data in the BIB background training set comes from a reliable expert and the prior policies can solve the task, we use this formulation for offline RL.

We use (state, action, next-state, reward) tuples to provide context to the model. These tuples are sampled from the familiarization trials in the episode. We engineer an artificial reward function based on the distance of the agent from the goal. For a state $s$, if the location of the agent is $x_a$, the location of the goal is $x_g$, then the reward $r$ is defined as:

$$r(s) = \begin{cases} -||x_a - x_g||_2 & \text{if } ||x_a - x_g||_2 < 20 \\ -100 & \text{if } ||x_a - x_g||_2 \geq 20 \end{cases} \tag{1}$$

**Model Description** The model has five components: state encoder; context encoder; Q-network (and the target Q-network); the policy network; and the prior policy network. The state encoder is pretrained similarly to the BC Model.

The context encoder is an MLP which takes (state, action, next-state, reward) tuples as input to produce a context embedding. The Q-network is an MLP that outputs the Q-value given the state, context, and action. The prior policy network and the RL policy network are identical MLPs that predict a 2-d Gaussian distribution in the action space. See table 7 for model specifications.

**Background Training.** The model is trained using ADAM optimizers with a learning rate of $5e - 4$ and default parameters. The model is successful on the background training tasks, but it is slightly worse than the BC models on the background tasks.

**Evaluation Tasks.** At test, we increase the number of transition context tuples used to infer the context to 100 (as opposed to 30 during training). The expectedness of an episode is measured by the likelihood of the observed actions in the test trial (see Table 8). The model does not improve on the behavior cloning model (similar to the effect observed in Siegel et al. (2020); the BM+MPO model is close to the BM prior when the demonstrations come from a reliable expert). Alternate

| PARAMETER | VALUE |
|---|---|
| CONTEXT MLP HIDDEN SIZES | [64, 64] |
| CONTEXT EMBEDDING SIZE | 32 |
| Q-NET MLP HIDDEN SIZES | [256, 128, 256] |
| POLICY MLP HIDDEN SIZES | [256, 128, 256] |
| PRIOR POLICY MLP HIDDEN SIZES | [256, 128, 256] |
| TARGET Q-NET UPDATE INTERVAL | 200 |
| GAMMA | 0.995 |

Table 7: Specifications of the Offline RL model.

offline-RL models might be able to improve on this model and use privileged reward information more effectively.

| BIB AGENCY TASK | OFFLINE RL |
|---|---|
| PREFERENCE | 45.7 |
| MULTI-AGENT | 52.2 |
| INACCESSIBLE GOAL | 42.8 [a] |
| EFFICIENCY: PATH CONTROL | 90.3 |
| EFFICIENCY: TIME CONTROL | 45.1 |
| EFFICIENCY: IRRATIONAL AGENT | 43.2 |
| EFFICIENT ACTION AVERAGE | 55.5 |
| INSTRUMENTAL: NO BARRIER | 92.4 |
| INSTRUMENTAL: INCONSEQUENTIAL BARRIER | 74.7 |
| INSTRUMENTAL: BLOCKING BARRIER | 37.9 |
| INSTRUMENTAL ACTION AVERAGE | 68.3 |

Table 8: Performance of the baseline Offline-RL model on BIB. The scores quantify pairwise VOE judgements.

[a]These results are from an older version of the task in which the expected and unexpected test trials were not perfectly matched (the location of the preferred and the nonpreferred objects was not matched). We predict that the Offline RL model's score would be higher on the current version but still lower than the BC model's score on this task.

# D   Comparing the performance of infants and AI systems

A potential challenge in comparing the performance of infants and AI systems on tests like BIB is that there is no metric that could suggest that infants' performance is 100%. So, what would it mean for an AI system to be as successful as infants on these tasks? Studies with infants focus on group-wise performance comparing looking times for expected and unexpected outcomes. When group-wise looking times are statistically different across outcomes, researchers infer that infants, in general, have certain expectations that reflect their knowledge about the world. When we designed BIB, we focused both on developmental findings that were well established in the field through multiple experiments and replications as well as on extensions of those findings, which would potentially inform new tests for infants.