# OpenReview forum: "Baby Intuitions Benchmark (BIB):  Discerning the goals, preferences, and actions of others"
_NeurIPS.cc/2021/Conference — NeurIPS 2021 Poster_

### Official Review · Reviewer_Yh1E · 2021-07-02

**Rating:** 6
**Confidence:** 3

**Summary:**


The paper starts from developmental studies on infants that showcase that infants have ways of ascribing certain types of intentionalities in what they observe, and proposes a benchmark dataset to evaluate whether modern AI/ML techniques are capable of the same type of reasoning. The paper presents a list of scenarios that demonstrate varying forms of intentionality, and also evaluates certain baseline AI/ML techniques on the benchmark scenarios to demonstrate the insufficiency of the baseline.

**Ethical Concerns:**

Not something obvious.

**Limitations And Societal Impact:**

Seems fine.

**Main Review:**

I will start with my conclusion:

Would I use the benchmark if it were available? Yes, in some form or another. At the very least as inspiration to develop new techniques within my area of interest, even if this might be different from what the authors have in mind (which seems to be deep learning over videos)

Is the paper "scientifically" optimal? No, neither in terms of presentation/motivation, nor in terms of convincing that the problem presented would end up with machines that have a notion of intentionality.

The authors mention the existence of another similar dataset recently released (which I did not check). I would expect a more elaborate discussion (not confined to a single paragraph) of the commonalities and differences.

Although I understand the underlying reason of why deep learning models are "expected" to fail the BIB tests, I am not sure whether a theory of "object preference" is needed for machines to do well in these tests. What is there to preclude an experimenter from biasing the deep learning in a manner that gives preferences to objects rather than location? Is there even a meaningful way to evaluate whether a deep learning architecture learns infant-like preferences, or ends up finding "shortcuts" to these tasks?

Since not all 8 familiarization instances are shown, it is hard to know what expectations one is expected to form in some of the presented scenarios. For example in the scenario of section 2.3, what in the familiarization instances should lead one to the expectation that if the preferred object is inaccessible, then reach for the other object? Shouldn't at least one such case be in the training data? Or are the two test instances shown simultaneously, and it is a matter of relative preference between the two?

I didn't quite understand the material offered. Is it several thousand (instead of 8) familiarization instances, and then several (instead of 1-2) test instances? Or is it several thousand episodes each comprising 8 familiarization instances and 1-2 test instances? In the latter case, is this really replicating the infant scenarios? I would say no, since the ML is learning also from the test instances (within the training episodes). This is clearly okay (since the evaluation episodes are not used), but it seems to change the setting from that in the infant developmental studies.

It is also unclear whether the trials (within each episode) show actual movement or just arrows indicating the movement. Figure 7 suggests the former, but up to this point in the paper the latter seems to be implied.

Some of my questions above are addressed in the appendix, but perhaps not so clearly in the main text. I think there is room to clarify things in the main text as well. I would propose that one of the scenarios (e.g., preference for target object over target location) is used throughout to demonstrate in more detail all the concepts, and then the rest of the scenarios can be presented at a higher-level. Also, section 3 seems to come rather late in the narrative. Perhaps the paper presented from the point of view of ML rather than developmental psychology would offer a more compelling narrative, given the target conference?

I am not sure if the 2D and 3D videos are necessary. Why not simply offer a symbolic representation (what you have in the JSON files) as the input?


**Time Spent Reviewing:**

3-4

---

> ### Author Response · Authors · 2021-08-10
> **Specific Response to @Yh1E**
>
> @Yh1E asked for clarification about the structure of the trials since not all 8 familiarization trials are shown in our figures, “It is hard to know what expectations one is expected to form in some of the presented scenarios. For example in the scenario of section 2.3, what in the familiarization instances should lead one to the expectation that if the preferred object is inaccessible, then reach for the other object? Shouldn't at least one such case be in the training data? Or are the two test instances shown simultaneously, and it is a matter of relative preference between the two?” @Yh1E rightly states that the two test instances are shown simultaneously, and it is a matter of relative preference between two carefully paired test trials (as are all of our evaluations). We will emphasize this in our revisions and provide clearer figures.
>
> @Yh1E asked, “Is it several thousand (instead of 8) familiarization instances, and then several (instead of 1-2) test instances? Or is it several thousand episodes each comprising 8 familiarization instances and 1-2 test instances? In the latter case, is this really replicating the infant scenarios? I would say no…” Thanks for these questions; neither of these interpretations is quite right. For background training, there are thousands of episodes each with 8 familiarization trials and just the one *expected* test trial. No unexpected trials are provided during background training. The tests themselves include 8 familiarization trials, one expected outcome, and one unexpected outcome. Tests of this structure are exactly what infants see in the lab. We think this might alleviate your concern, but please let us know if it does not (See also discussion at beginning of Section 3 on the role of background training).
>
> @Yh1E also asks why we did not use a symbolic representation as the input. We tried baselines that directly used symbolic representations as input but they did not offer any improvements over the other baselines (which rely on visual inputs). We would be happy to include these baselines in the appendix as well, or at least a mention of them, if @Yh1E sees this as an important addition.
>
> We thank @Yh1E for their comments on the clarity of the paper and will work to improve the figures to show that there is actual motion of the agent in the scene.
>
> Kindly see our “General response” for our response to other issues that were raised.

---

### Official Review · Reviewer_uke7 · 2021-07-15

**Rating:** 6
**Confidence:** 4

**Summary:**

This paper presents the Baby Intuitions Benchmark (BIB) to test deep learning methods abilities’ to reason about how other agents will behave. BIB is inspired by human infants’ expectations about other agents, as clearly laid out in the paper. The authors include several different capabilities/tasks within the benchmark along with their developmental background, testing 3 different models across 5 different tasks. These models only perform well in the simplest of tasks, suggesting that the benchmark tests capabilities that are not trivial to acquire with current methods.

**Limitations And Societal Impact:**

I see no obvious limitations/societal impact with the proposed benchmark.

**Main Review:**

To summarize my review, I found the paper compelling as a pointer to how rich human social inferences can be, and just how lacking a variety of ML baselines are in this respect. My most critical technical comments relate to including a model that relies on bayesian inference, as motivation for how well models could perform if they had the right kinds of structure in them (as in AGENT [1]). While I am not sure how much this benchmark will be picked up by the general machine learning community, it does point out specific limitations with current methods for social inference, and could therefore be useful especially for researchers in human computer interaction, human robot interaction, and multi-agent ML researchers generally.

However, I am choosing a low rating because this is not the appropriate track for this paper. This paper does not make a meaningful contribution for the main NeurIPS track, in that it does not show new results either for cognitive science/neuroscience, or provide new algorithms for machine learning methods. It instead tests a variety of existing methods on a new benchmark. It should absolutely be submitted to the “Datasets and Benchmarks” track. If it is possible to move the paper to this other track, my rating would jump to 7 for that track. This alternative track has a deadline for submission of August 27, please see https://neurips.cc/Conferences/2021/CallForDatasetsBenchmarks for details.

For context, I think that for a benchmark to be accepted to the main track, it needs to either (1) provide access to data that would otherwise be difficult for individual researchers to collect, like neural recordings, human annotations, or human experimental data, or (2) clearly address an important ML problem, and provide clear reasons for why the benchmark will be very widely taken up within the ML community. While I like the BIB benchmark, it does not fit into (1), and I am not convinced that it clearly falls into category (2) either (due to similarities to existing benchmarks, as well as BIB’s somewhat bespoke passive setting where agents cannot interact with the environment), which is why I strongly recommend considering the datasets and benchmarks track.

### Excellent paper layout - very clear writing

The section headings - stating the question of the specific sub-task - and then giving the developmental background and the specifics of the familiarization and test trials, was very clear. My only additional suggestion would be to include a small wrapped figure in each of these sections (like Figures 4, 5, and 6) to help a reader visualize what the task looks like as well.

### Reasonable ML baselines, although could include bayesian inference agent

The baselines included by the authors, an offline RL, behavioral cloning, and video RNN models, seemed reasonable for the presented task to cover a variety of approaches that ML researchers might try first. The authors explained very well why inverse reinforcement learning approaches are not suitable for this benchmark, precisely because agents do not have access to the underlying environment to attempt to create their own plans.

My only suggestion would be to include an “aspirational” baseline as in AGENT [1], that uses significantly more structure (in the form of objects and agents), but as a result is able to solve the presented tasks. I think this could help convince readers that the benchmark is interesting, as it pushes on building these more structured representations in order to enable success. Without something like this, researchers might think the benchmark is just too difficult, and not try it.

### Benchmark's passive nature may prevent widespread adoption

The benchmark's passive nature is a limitation in that most real setups with human-computer interaction, for example, do give agents access to the environment, and this helps dramatically with tasks like goal inference and inverse reinforcement learning. As a result, I think this benchmark as proposed will be slightly less relevant for the wider swath of the NeurIPS community, although it is well motivated in the context of developmental cognitive science.

### Behavioral validation incomplete - do infants really perform at 100% accuracy for these problems?

A lingering question I had throughout looking at the paper was: how well would infants really perform on these tasks? As I understand the developmental literature for these kinds of tasks, it is generally not the case that 100% of infants choose the correct stimulus 100% of the time. Can the authors comment on this? Is there a success percentage less than 100 that would be sufficient to say that a machine learning model has performed at the same level as an infant for some of these different tasks?

### Questions about benchmark size

The authors use a breakdown of 10,000, 10,000, 4000, and 4000 tasks in the “background” training. Can the authors please clarify why they used this particular breakdown?

[1] Shu et al, ICML 2021. AGENT: A Benchmark for Core Psychological Reasoning

**Time Spent Reviewing:**

3

---

> ### Author Response · Authors · 2021-08-10
> **Specific Response to @uke7**
>
> @uke7 said that they would have scored our paper as a 7, rather than the current 5, if we had instead submitted it to the datasets and benchmark track. The call for papers made it clear that datasets and benchmarks were welcome through either process. We saw our contribution as fitting squarely at the intersection of cognitive science and machine learning, so we submitted to the main conference where cognitive science is an explicit subject area and where we could receive feedback from and disseminate to researchers with expertise and interest in cognitive science and machine learning. Beyond BIB’s contribution as a test of agency reasoning, we see our approach in designing BIB as a framework for converting familiarization-test based experiments adopted from research in cognitive development to evaluate machine learning approaches. As @dLBP puts it, “[BIB] tackles one of the most critical topics in developmental psychology from a machine learning perspective”, and @KaGu points out that “the cognitive science perspective is also critical for motivating a new paradigm for evaluating artificial agents”. With these points considered, we sincerely hope that @uke7 will see value in our work appearing on the main track and reconsider the deduction of points based on this issue.
>
> @uke7 rightly pointed out that infants do not always perform at 100% on lab tasks and asked what this implies for tests of machine learning models. Studies with infants focus on group-wise performance comparing looking times to unexpected and expected outcomes. When group-wise looking-time performance is statistically different across outcomes, researchers infer that infants, in general (not an individual), have such expectations. Inspiration from these inferences for tests of ML is thus challenging! So when we designed BIB, we considered and adapted only developmental findings that were well established in the field through multiple experiments and had been replicated both within and across laboratories. For example, our Preference test was inspired by a seminal paper in the field of infant cognition, cited more than 2000 times. We will better highlight these points in our revision.
>
> @uke7 pointed out that the behavioral validation of our benchmark is incomplete. While the stimuli are directly inspired by developmental psychology, we are also actively conducting studies to measure infants’ performance on BIB, as there are some interesting differences between the way infant stimuli are typically displayed and the way BIB’s stimuli are displayed. The sheer potential for these exact stimuli to be used directly with infants is, to our knowledge, entirely novel, and we are excited about the direct dialogue that this possibility promises to create a bridge between cognitive science and ML/AI. Because of the time and resources associated with conducting behavioral experiments with infants, however, we believe a full validation is outside the scope of the present paper.
>
> @uke7 suggests that the “benchmark's passive nature may prevent widespread adoption”. In BIB, a model cannot actively sample from the environment and can only use the samples provided in the dataset. This is an essential feature because many models operating in real-world scenarios have very limited access to privileged information in the form of active interactions (e.g. they are third-party observers of other people’s actions). As we point out in the paper, but will highlight in our revision, infants rely on little to no active interaction with a particular environment to make meaningful inferences and predictions about the agents in that environment, and infants’ inferences are far more abstract than their particular observed or active experience [1, 2, 3, 4].
>
> For the background training, we provide a sufficient number of videos so that the model can be highly successful on generalizing within distribution on the background training tasks.
>
> Kindly see our “General response” for our response to other issues that were raised.
>
> [1] Skerry, A. E., Carey, S. E., & Spelke, E. S. (2013). First-person action experience reveals sensitivity to action efficiency in prereaching infants. Proceedings of the National Academy of Sciences, 110(46), 18728-18733.
>
> [2]Liu, S., Brooks, N. B., & Spelke, E. S. (2019). Origins of the concepts cause, cost, and goal in prereaching infants. Proceedings of the National Academy of Sciences, 116(36), 17747-17752.
>
> [3] Gergely, G., Bekkering, H., & Király, I. (2002). Rational imitation in preverbal infants. Nature, 415(6873), 755-755.
>
> [4] Zmyj, Norbert, Moritz M. Daum, and Gisa Aschersleben. "The development of rational imitation in 9‐and 12‐month‐old infants." Infancy 14.1 (2009): 131-141

---

> > ### Comment · Reviewer_uke7 · 2021-08-21
> > **Thank you for your reply!**
> >
> > Thank you to the authors for their considered and detailed response. I am glad to hear that there is ongoing work to determine the behavior of infants on BIB, that sounds extremely exciting indeed!
> >
> > Given (1) the existence of AGENT, which was concurrently developed with the current paper, (2) the authors' points about wishing to have Cognitive Science reviewers instead of general ML reviewers, which I can definitely sympathize with, and (3) the fact that the current set of reviewers is not at all guaranteed to be reassigned to this paper if it was moved to datasets/benchmarks, I am raising my score to a 6. Even though I believe this paper belongs in datasets/benchmarks, it still deserves to be published!

---

### Official Review · Reviewer_KaGu · 2021-07-15

**Rating:** 7
**Confidence:** 3

**Summary:**

EDIT AFTER AUTHOR RESPONSE:  One concern I originally had was whether researchers would actually find this dataset useful, but the review by @Yh1E and @uke7 improved my impression of the utility of the dataset.  I am also pleased that the authors will comment more on the relation of their problem setup to classical MDP and POMDP frameworks.  Hence I am raising my score from 5 to 7.

This paper asks whether AI systems can demonstrate similar theory of mind (ToM) capabilities as human infants.  It identifies a number of key ToM capabilities that have been established in the cognitive science literature, such as learning preferences of observed agents, and predicting that agents will take the most efficient path to their goal.  The paper introduces a benchmark dataset with 2d and 3d videos with simulated agents for the purpose of testing ToM capabilities in AI systems.  Each task in the dataset consists of a familiarization phase with videos intended to allow learning of agent behavior, and then a test phase where the agent exhibits either expected or unexpected behavior.  Taking inspiration from the fact that human babies look longer at unexpected stumli, the paper introduces a "violation of expectations (VoE)" framework for evaluating AI systems.  Under this paradigm, an AI system is considered to demonstrate evidence of human-like learning if it has greater prediction error for video frames in the unexpected condition than in the expected condition for the testing episode.  It evaluates two recent approaches on this dataset, behavior cloning, and offline RL.

**Limitations And Societal Impact:**

The paper does not adequately address the limitations of the benchmark, as there is no attempt to rule out alternative means that artificial agents without ToM capabilities might still be able to pass the BIB benchmarks.  The performance of benchmark methods is surprisingly bad and needs to be explained.  It is not clear that BIB dataset has enough or sufficiently varied training data to allow for generalization.

The paper could be improved by further explanation of the performance of the methods that were evaluated, addressing the adequacy of the amount and diversity of training data, and the paper should also spend some effort attempting to rule out potential alternative explanations for a method to pass the BIB benchmark.  The paper could also discuss the extent to which the VOE paradigm adds to the existing state-of-the-art in evaluating artificial agents, beyond the naturalistic argument of attempting to match human behavior.  After all, one does not have to imitate human behavior to achieve human-level performance, as human-like performance may plausibly emerge as a by-product of trying to optimize a simpler criterion.

**Main Review:**

Originality:  High.  The paper takes a very different approach to the problem than comparable recent works in RL.  It doesn't start with any formal theory of how we might model a real-world agent (e.g. POMDP) but rather takes the cognitive science literature as a starting point.  The strong grounding in human cognitive science is a strength of this work.  The cognitive science perspective is also critical for motivating a new paradigm for evaluating artificial agents: testing whether artificial agents experience similar prediction errors to humans when seeing agents demonstrate behavior that is inconsistent with previously observed behavior.  That said, it is not clear the degree to which VOE adds to agent evaluation beyond correct prediction. If an artificial agent has high predictive accuracy, doesn't it also pass the VOE benchmark?  How does VOE add non-redundant information about the ToM capabilities of an artificial agent?

Quality:  Mixed.  For a psychology venue, this would be a great paper.  But relative to the ML literature, the technical sophistication here is lacking.  The paper starts off with the cognitive science literature, which is great, but it does not go further to completely bridge to the formalisms that have been used in the artificial intelligence literature.  The technical contribution of this paper would be elevated if the paper made a serious effort to consider what human learning capabilities say about the kinds of agent models humans are likely to employ.  For instance, is there a particular type of POMDP model that would result in an agent that has stable preferences, which takes instrumental action to reach goals, and which takes the path of least effort to get there?  Another potential contribution that was not addressed by this paper is to think about how formalisms used in AI could be used to better understand infant cognition.  To what extent are infants able to learn to predict general POMDP agents?  The results also cast doubt on the correctness of the evaluation.  In the Shu et al. paper, the BC method was able to achieve above-chance prediction on the goal preference task, so why does it get below-chance performance in the BIB dataset?  How do we know that the BIB dataset has sufficient training data to enable performance comparable to human infants?

Clarity:  The paper is clearly written, although the VOE testing criteria could be emphasized more as it is rather novel.

Significance:  Moderate.  It seems valuable to look at human behavior to motivate benchmarks for AI, but the benchmark introduced here still seems rather limited.  Even if an artificial agent could be developed that would pass all of the tasks in the benchmark, I am not convinced that this would be conclusive evidence for the agent to have strong theory of mind.  The fact that BC and IRL perform poorly could be an important finding, but it depends on the reason for their failure, which has not been conclusively established here.

Other comments:
* Supplemental materials - some videos are mislabeled (expected switched with unexpected).  I could not play some of the videos (e.g. eval-videos-3d/efficient_action_b_expected.mp4).
* Line 322: "characteristics"

**Time Spent Reviewing:**

4

---

> ### Author Response · Authors · 2021-08-10
> **Specific Response to @KaGu**
>
> @KaGu suggested that we could have gone further to bridge the gap between AI and cognitive development by connecting with standard formalisms in RL. We appreciate this suggestion, and we will provide additional details for how the BIB environment used to create the videos can be seen as an MDP in terms of the state and action space, the transition functions, etc.
>
> Work in computational cognitive science has utilized POMDPs in Bayesian theory of mind models of agency reasoning, and we will update our paper to include a longer discussion of the advantages and challenges of using such an approach for BIB, which is larger-scale and noisier than previous applications such as those used in Baker et al. [1, 2, 3].
>
> @KaGu also suggested that the benchmark was limited insofar as if a model passed it, they would not be convinced that the model had a “strong theory of mind.” We take @KaGu’s reason for this comment as the possibility of the model taking shortcuts, which we address in our “General response.” We also note that BIB focuses on one of two proposed core components of human theory of mind, i.e., understanding the *intentions* of other agents. To achieve a strong theory of mind, a model would not only have to understand intentionality, it would also have to understand the phenomenality of other agents, e.g., what they can see or what they believe. These two core cognitive abilities together (intentionality and phenomenality) are thought to underlie human theory of mind abilities and lead to, for example, success in a false belief task in which a human would have expectations about where an agent would search for a goal object based on where that agent last saw it. We will flesh out this framing in our revision, and we plan to address the idea of phenomenality in our future work.
>
> Kindly see our “General response” for our response to @KaGu’s other issues that were raised, including questions about shortcuts and baseline performance.
>
> [1] Baker, C. L., Saxe, R., and Tenenbaum, J. B. (2009). Action understanding as inverse planning. Cognition, 113(3):329–349.
>
> [2] Baker, C., Saxe, R., and Tenenbaum, J. (2011). Bayesian theory of mind: Modeling joint belief-desire attribution. In Proceedings of the annual meeting of the cognitive science society, volume 33.
>
> [3] Baker, C. L., Jara-Ettinger, J., Saxe, R., and Tenenbaum, J. B. (2017). Rational quantitative attribution of beliefs, desires and percepts in human mentalizing. Nature Human Behaviour, 1(4):1–10.

---

### Official Review · Reviewer_dLBP · 2021-07-17

**Rating:** 7
**Confidence:** 5

**Summary:**

Directly inspired by the literature on infant cognition, the authors propose a benchmark to challenge machines' social cognition with a violation of expectation paradigm.

**Limitations And Societal Impact:**

See main review

**Main Review:**

+ This paper is exceptionally well-motivated and tackles one of the most critical topics in developmental psychology from a machine learning perspective. It provides reasonable problem setups.
+ The experiments and protocols are appropriately chosen. The experiments were selected from classic scenarios in developmental psychology. The VOE protocol is also basically the standard in the field.
- Although the experiments replicate the classic studies in developmental, and the questions proposed in the paper are valid, how could the experiments and the study designs prevent the learning system to simply convert this problem into a simple classification problem, therefore may not eventually possess the abilities that the authors hope to give to the machines.
- Of note, this paper was submitted to ICML and almost made it; I was one of the reviewers. The long meta-review commented that all reviewers find the paper interesting but lack proper baseline. In the current revision, the authors did add an additional baseline.

**Time Spent Reviewing:**

1

---

> ### Author Response · Authors · 2021-08-10
> **Specific Response to @dLBP**
>
> @dLBP suggested that the experimental designs prevent the model from simply converting the task into a classification problem, as we provide only the expected examples in the background training (infants are also not privy to a larger number of “unexpected trials,” which would be needed to train a supervised classifier).
>
> Kindly see our “General response” for our response to other issues that were raised.

---

### Author Response · Authors · 2021-08-10
**General Response**

We thank the reviewers for their careful reading and consideration of our work. We are pleased that all of the reviewers spoke positively about its contribution.  For example, @dLBP found our work to be “exceptionally well-motivated” in tackling “one of the most critical topics in developmental psychology from a machine learning perspective.” @KaGu commented that our inspiration from developmental cognitive science motivates a “new paradigm for evaluating artificial agents.” @uke7 suggested that BIB is “compelling as a pointer to how rich human social inferences can be, and just how lacking a variety of ML baselines are in this respect.”  @Yh1E remarked that BIB serves “as inspiration to develop new techniques within my [their] area of interest.” The reviewers nevertheless also raised some important queries and critiques. We thank the reviewers for their comments, and in our response, we address the reviewers’ general concerns, and we reply individually to the reviewers’ specific comments. We believe that addressing all of these comments in a revision will make our paper stronger.

@KaGu, @uke7, and @Yh1E asked for more information on the similarities and differences between BIB and AGENT, a benchmark developed contemporaneously to BIB by Shu et al. (2021). We will expand on this topic in our revision, but here are some key distinguishing features. BIB and AGENT cover different competencies informed by different developmental findings. For example, AGENT does not evaluate instrumental actions and scenarios with multiple agents; BIB does. Moreover, AGENT trains and tests on the same types of scenarios: It examines generalization by dropping a few scenarios from training and observing the performance on these held out scenarios. In contrast, each of BIB’s tests is systematically different from the background training scenarios. For example, to solve BIB’s Preference Test, models need to flexibly combine learning from two types of background training scenarios (no-preference navigation trials and no-navigation preference trials). These two differences between BIB and AGENT, in particular, are important because they target the generalizability and abstractness of a model’s representations of other agents.

The differences between AGENT and BIB are also related to @KaGu's and @Yh1E's queries about baselines, and so again we are happy to expand on this topic in a revision. @KaGu asked why behavior cloning (BC) failed on BIB while performing above chance on AGENT. BC and offline-RL are known to perform poorly on out-of-distribution and systematically different scenarios from training to test. Because systematic generalization is a primary feature of BIB but not AGENT, this is likely why BC models perform worse on BIB. @Yh1E asked why we did not have a Bayesian inference baseline similar to AGENT. Extending a structured Bayesian approach to BIB entails handcrafting several domain-specific features which is not straightforward, although we hope this approach will be explored in the future.

@KaGu asked why the violation of expectation (VOE) metric was chosen over predictive accuracy. While there are similarities between these two metrics, they are importantly different for our purposes. VOE allows us to focus on theoretically interesting alternatives and lures that would fool simpler models and heuristics. The paired unexpected and expected trials differ at one or two critical junctures where abstract and human-like reasoning happens (e.g., forming a preference for a goal object instead of a goal location). Thus, a model that misses the critical juncture could still have high overall accuracy. The very same idea goes into the design of effective studies on infant cognition.

Along these same lines, @KaGu and @Yh1E asked if the design of the evaluations prevents a model from finding shortcuts. We carefully considered this issue and adopted contrasts directly from the infant cognition literature, which is challenged by the very same question (i.e., are infants solving these problems using abstract conceptions or perceptual shortcuts)!  The unexpected scenarios were designed to lure models that rely on shortcuts: the lures are more perceptually similar—along various dimensions—to the familiarization and background training trials. Further protection is provided by the fact that only expected trials are given for background training (see the response to @dLBP). Our revised manuscript will further highlight how specific contrasts in our unexpected and expected trials address possible shortcuts.

-- Summary --

Reasoning about other agents is a core component of human intelligence that is comparatively underdeveloped in machines. As a community of researchers in machine learning and AI, if we want to build more human-centered AI (or more human-like AI) that interacts well with humans, BIB — and other benchmarks inspired by BIB — will be critical in moving the field forward. In sum, we believe that BIB provides more than just a benchmark test of reasoning about other agents. In the words of @KaGu, it provides a “new paradigm,” for translating tests of human intelligence into those for machine intelligence.

---

> ### Comment · Reviewer_KaGu · 2021-08-12
> **Discussion on novelty of VOE**
>
> > @KaGu asked why the violation of expectation (VOE) metric was chosen over predictive accuracy. While there are similarities between these two metrics, they are importantly different for our purposes. VOE allows us to focus on theoretically interesting alternatives and lures that would fool simpler models and heuristics. The paired unexpected and expected trials differ at one or two critical junctures where abstract and human-like reasoning happens (e.g., forming a preference for a goal object instead of a goal location). Thus, a model that misses the critical juncture could still have high overall accuracy. The very same idea goes into the design of effective studies on infant cognition.
>
> That makes sense to me.  It seems, then, that the key contribution of the VOE paradigm is the pairing of two (or more) potential outcomes, which may not differ much in terms of classical prediction error metrics, but which do differ in some experimenter-defined way, which is what you call a "critical juncture" here.  The value of VOE, therefore, is that if we want to understand some specific capability of AI systems, we can come up with a customized test for checking that capability by comparing two close alternatives which humans can only distinguish by relying on that specific capability.  Would you agree with this framing of the VOE paradigm?
>
> For me, I found the specific reliance on comparing prediction errors between the options to be somewhat unprincipled, but if we view the VOE paradigm in the more general way that I outlined, then I think the paradigm becomes much more promising as a general tool, because rather than using prediction error comparisons as was done in this paper, which seems rather less than ideal due to the arbitrariness of the error metric, actually any method for using a trained model to select the best outcome out of a set of alternatives can be used to assess VOE performance.

---

> > ### Author Response · Authors · 2021-08-12
> > **RE: Discussion on novelty of VOE**
> >
> > Yes; you put it beautifully! We will certainly incorporate this point into the revision. Thank you!

---

### Decision · Program_Chairs · 2021-09-27

**Decision:**

Accept (Poster)

**Comment:**

This paper introduces a new benchmark set for visual cognition tasks, inspired by looking-time experiments done with human babies. I think this work would be useful even if the only role were to highlight the questions and methodologies used by developmental psychology to student non-verbal intelligence, but care was taken to provide a broad benchmark that I think will provoke interesting ML work. The reviewers generally agree, though note a few requests for clarification and context.

(Note, it's probably the case that this paper fits better into the D&B track, but since that track is brand new this year I don't think the authors should be penalized for choosing the main track.)